# Sample Size Estimation for Chest X-ray Classification with Foundation Models

## Abstract

The integration of deep learning models into clinical practice, particularly in radiology, is often hindered by the need for large, meticulously labeled datasets, which entails significant time and financial costs. While foundation models substantially reduce this dependency, a critical question remains: what is the minimum amount of annotated data sufficient to achieve clinically acceptable accuracy? In this work, we introduce a methodology for accurately predicting sample size requirements by modeling learning curves with a power law. Our study demonstrates that modern foundation models, such as XrayCLIP and XraySigLIP, not only outperform traditional architectures but also achieve high ROC-AUC scores with significantly fewer training examples. A key finding of our research is the evidence that the learning dynamics observed with a sample of just 50 labeled cases can predict the model's asymptotic performance with high precision. Thus, our study offers a scientifically grounded approach to optimizing the data annotation process, enabling researchers and clinicians to minimize costs and accelerate the development of reliable diagnostic tools.

## 1 Introduction

While significant literature exists on deep learning methods for chest X-ray classification (Meedeniya et al., 2022), comparatively little attention has been paid to efficient training size estimation in this context (Viering & Loog, 2022). Recent progress in foundation models has further heightened the importance of this question: not only can these models achieve higher accuracy, but their learning curves may also be more predictable with fewer labeled samples.

A key principle of this study is the rigorous and unbiased evaluation of labeling efficiency. Although the MIMIC-CXR dataset contains structured labels for some pathologies, we confirm that none of the evaluated foundation models—RAD-DINO-MAIRA-2, XrayCLIP, and XraySigLIP—were pretrained using these specific structured annotations. The RAD-DINO-MAIRA-2 model was trained solely on images, while XrayCLIP and XraySigLIP were trained on image-text pairs using unstructured, free-form reports. This methodological choice is crucial as it ensures a level playing field for all pathologies in terms of the models' prior knowledge, allowing for a fair assessment of their sample efficiency in a real-world fine-tuning scenario.

Motivated by these considerations, we propose a systematic approach to estimate how many annotated examples a given model requires to meet a clinically relevant ROC-AUC threshold, leveraging power-law fitting of the learning curves.

## 2 Related work

Chest X-ray is a crucial diagnostic imaging modality that provides rapid and cost-effective insights into various pulmonary and cardiac conditions (Raoof et al., 2012). The classification of chest X-ray pathologies is well-studied, and training machine learning models for new conditions is relatively straightforward, although it still relies on large annotated datasets to achieve clinically acceptable accuracy (Çallı et al., 2021).

Recently, general self-supervised learning (SSL) frameworks such as DINO (Oquab et al., 2023) and CLIP (Radford et al., 2021) have shown great promise for imaging tasks by learning robust feature

representations from massive unlabeled datasets. These frameworks differ in their training objectives: DINO is purely image-based self-supervision, whereas CLIP leverages paired text–image data for multi-modal alignment. Building on these advances, specialized chest X-ray foundation models (*e.g.*, RadDINO (Pérez-García et al., 2024), XraySigLIP/XrayCLIP (Chen et al., 2024)) adapt these frameworks to chest x-ray imaging.

Beyond predicting performance scaling, another prominent approach to data-efficient learning is to improve the quality of the learned representations themselves. For instance, Supervised Contrastive Learning (SupCon) has emerged as a powerful technique that goes beyond the standard cross-entropy loss function (Moradinasab et al., 2024). SupCon aims to pull representations of examples from the same class closer together in the latent space while pushing apart examples from different classes (Wu et al., 2023). This approach has shown significant promise in medical imaging, where it can help learn more discriminative features even with limited labeled data. However, SupCon can face challenges such as "class collapse", where intra-class variance is lost (Chen et al., 2022), and its effectiveness can diminish under severe class imbalance, a common scenario in medical diagnostics. While these methods aim to fundamentally alter the training objective to learn a *better* representation space, our work addresses the complementary, pragmatic question: for widely-used, off-the-shelf foundation models, how can one reliably predict their performance trajectory to inform annotation budgets? Thus, our study focuses on the practical estimation of sample size requirements rather than the development of new representation learning techniques.

Though no works have directly explored learning curve estimates for chest X-ray classification tasks, many investigations in other domains (*e.g.*, machine translation, image recognition, and speech recognition) rely on power-law approximations to characterize how performance improves as the training set size grows (Cortes et al., 1993; Gu et al., 2001; Hestness et al., 2017). Nonetheless, numerous studies reveal that learning curves can be well-behaved or ill-behaved, with phenomena such as double descent and peaking complicating straightforward sample-size extrapolation (Raudys & Duin, 1998; Devroye et al., 2013; Nakkiran et al., 2020). A popular strategy is to estimate the asymptotic accuracy by measuring the early slope of a power-law fit and extrapolating the eventual plateau in performance (Hoiem et al., 2021; Frey & Fisher, 1999; Kolachina et al., 2012). Additionally, a relevant idea is to incorporate progressive sampling, dynamically refining the power-law estimate of the learning curve so as to reduce annotation overhead (Provost et al., 1999).

## 3 METHODS

### 3.1 DATASET CONSTRUCTION

A popular open dataset MIMIC-CXR (Johnson et al., 2019) was used as the data source for this study. Using RadGraph annotations (Jain et al., 2021), we extracted structured "organ–pathology" labels. These labels underwent a normalization to merge synonymous anatomical terms into unified categories (*e.g.*, unifying the tokens "lung" and "lobe") and then split into two groups: normal and pathological. Pathological annotations were further clustered according to their common pathogenetic mechanisms to reduce redundancy. In total, 21 distinct pathology classes where selected. An example of chest X-ray, corresponding RadGraph findings, and selected pathologies are shown in Figure 1.

For each resulting pathology, we created a binary classification dataset, where a confirmed pathology was labeled as a positive class. The negative class consisted of studies corresponding to a normal anatomical-physiological state of the target organ, in a 1:5 ratio. If for some classes negative examples were insufficient, existing data were duplicated to maintain the balance.

Each pathology-specific dataset was split into training, validation, and test subsets using a deterministic method with a fixed seed value. The validation and hold-out test subsets were each assigned 10% of the total data in a stratified manner, preserving the 1:5 class ratio. The remaining 80% made up the full training pool.

The choice of fixed increments for positive cases (ranging from 5 to 1000 samples) and a constant 1:5 class ratio was a deliberate methodological decision. This controlled environment, while an abstraction of clinical prevalence, was necessary to isolate the effect of the number of positive

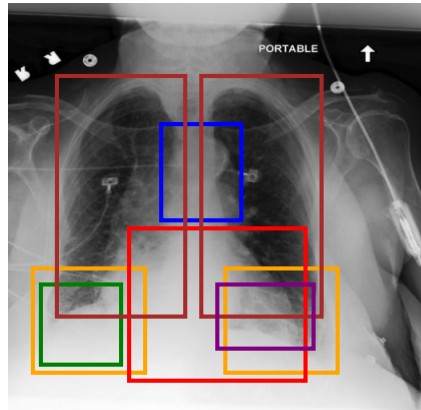

Hiatal hernia

Bibasilar atelectasis consolidation

pleural effusion

aorta calcified tortuous

cardiac enlarged

lungs hyperinflated

Figure 1: A chest X-ray with pathology labels extracted via RadGraph.

examples on performance and to ensure a systematic, comparable analysis across a wide range of pathologies and their varying data availability.

In each experiment below, the training sets were formed for a feature-based transfer learning regime as follows:

1. The number of positively annotated samples was purposely restricted with a value $N_{\text{cases}}$ taken from the set $\{5, 10, 15, ..., 45, 50, 100, 250, 500, 1000\}$.

2. From the initial training pool, $N_{cases}$ pathological studies were randomly selected (using a unique seed for each experiment).

3. Negative cases were added in a 1:5 ratio to preserve the original balance.

The validation and the testing sets were the same for each pathology in all experiments.

## 3.2 MODELS TESTED

For feature extraction, we employed 3 different chest x-ray foundation models. The first, RAD-DINO-MAIRA-2 (Bannur et al., 2024), is a transformer pre-trained using the DINOv2 framework on a heterogeneous corpus of 1.2 million medical images. The second and the third are XraySigLIP and XrayCLIP – also transformer models, but pre-trained using the CLIP (Radford et al., 2021) and SigLIP (Zhai et al., 2023) frameworks, respectively, on a million image-text pairs from CheXinstruct (Chen et al., 2024) dataset. To verify the robustness of the results and to establish a baseline, we also used ResNet-50 convolutional neural network (He et al., 2015) pre-trained on ImageNet as a baseline encoder. For each of the tested models we constructed the classifier head by applying a dropout layer (p = 0.1) to the encoder's pooled features, followed by a linear projection to a single output unit.

## 3.3 TRAINING PROCEDURE

Training was done using the `transformers` library (PyTorch backend) with the AdamW optimizer (binary cross-entropy loss, an initial learning rate $2 \times 10^{-5}$), a cosine annealing learning rate scheduler without warm-up, a batch size of 64, and an early stopping after 4 consecutive epochs without improvement in the validation loss. During training, the image encoder weights were frozen to retain their pre-trained representations, and only the linear binary classifier head was trained.

We applied train augmentations combining geometric and photometric modifications: a horizontal flip (50% probability), affine transformations with a random rotation between $-90°$ and $90°$ and a rotation center is center of the image size, photometric adjustments via linear brightness adaptation

within $\pm 35\%$ of the original values alongside non-linear gamma correction of contrast in the same range, and spatial cropping with random square crops, covering 3–33% of the original image size.

### 3.4 Power law fitting

To model the scaling behavior of the classifier, we fit a power law function to the area under the receiver-operating characteristic curve (ROC-AUC) in the following form:

$$\text{ROC\_AUC}(n) = \alpha - \frac{\beta}{n^{\gamma}}, \tag{1}$$

where $n$ is the number of distinct positive examples in the training set, $\alpha$ represents the asymptotic performance, $\beta$ controls the deviation from the asymptote, and $\gamma$ governs the rate of convergence. Several curve-fitting approaches (linear, exponential, and power-law) were considered. The three-parameter power-law function, shown in Equation 1, was selected due to its consistently superior fit to the empirical learning curves across the range of pathologies and models evaluated, consistent with prior research on scaling laws in deep learning (Hestness et al., 2017; Kaplan et al., 2020; Viering & Loog, 2022).

To estimate the parameters $\alpha$, $\beta$, and $\gamma$, we employ non-linear least squares fitting using the `curve_fit` function from `scipy.optimize`. For the fitting procedure, we specify an initial guess and bounds for the parameters to ensure reasonable behavior of the model. In our case, we set the initial guess as $\alpha = 0.95, \beta = 0.5, \gamma = 1.0$ with the following bounds: $\alpha \in [0.8, 1.0]$ ensuring the asymptotic value is near 1, $\beta \geq 0$ to maintain non-negative deviation, $\gamma \geq 0$ for a proper convergence rate.

For the fitted power law we use the following notation $\text{ROC\_AUC}_{N_{\text{cases}}}(n)$ where $N_{\text{cases}}$ is the maximum number of examples used to fit the curve. For example, $\text{ROC\_AUC}_{20}(n)$ stands for the power law curve, fitted on the experimental data points $N_{\text{cases}} = 5, 10, 15, 20$. Finally, given the fitted curve, we draw a conclusion about the optimal number of required labeled samples $n_o$ by evaluating where the curve starts exceeding a certain clinically-relevant threshold ($\text{ROC\_AUC}_{N_{\text{cases}}}(n_o) = 90\%$)[1].

## 4 Results

### 4.1 All pathologies data points

The results for all patologies are presented in Table 1 . For the 4 models and each of the pathologies we provide the experimental ROC-AUC on all the training data avalable for this pathology, and the expected number of cases needed to reach ROC-AUC 0.9. This number was calculated by fitting a power law to all the experimental data using less than 50 training samples and using it to calculate the number of examples needed.

### 4.2 Number of training examples vs experimental ROC-AUC

Figure 2 illustrates the core principle of our methodology using the "lobe mass" pathology as an example. It demonstrates that power-law curves fitted on a small fraction of the data (*e.g.*, ROC-AUC$_{50}$, fitted on $N \leq 50$ cases) closely track the true empirical learning curve (ROC-AUC $\pm$ 1 Std). This visual evidence supports our hypothesis that early learning dynamics are highly predictive of future performance, forming the basis for reliable sample size estimation.

### 4.3 Comparison of Early Slope and Final Performance

To quantify the relationship between early-stage learning behavior and final performance, we calculated the Pearson correlation coefficient ($r$) between the initial slope of the learning curve and

---

[1]The ROC-AUC threshold of 0.90 used throughout this study was selected as an illustrative benchmark for simplicity and consistency. However, our methodology is generalizable and can readily accommodate any clinically relevant performance threshold, allowing practitioners to adjust the labeling requirements according to specific diagnostic standards.

Table 1: Performance metrics with best values highlighted in bold. Best ROC-AUC and best n@90 are shown in bold.

| Pathology | ResNet-50 roc | ResNet-50 n@90 | RAD-DINO roc | RAD-DINO n@90 | XrayCLIP roc | XrayCLIP n@90 | XraySigLIP roc | XraySigLIP n@90 |
|---|---|---|---|---|---|---|---|---|
| pulmonary_fibrosis | 0.85 | 2545 | 0.92 | 104 | 0.97 | 24 | **0.99** | **8** |
| pericardial_effusion | 0.65 | >1M | 0.73 | 98922 | 0.77 | 5486 | **0.92** | **79** |
| aortic_dissection | 0.72 | >1M | 0.81 | 241 | 0.71 | 4656 | **0.98** | **18** |
| hiatal_hernia | 0.78 | >1M | 0.92 | 646 | 0.90 | 317 | **0.93** | **120** |
| lobe_mass | 0.71 | >1M | 0.81 | 156K | 0.86 | 7605 | **0.96** | **53** |
| hemidiaphragm_eventration | 0.78 | 1805 | 0.84 | 5315 | **0.86** | **688** | 0.84 | 8954 |
| fissure_fluid | 0.64 | inf | 0.67 | 162K | 0.75 | 246K | **0.95** | **45** |
| spine_deformities | 0.81 | 1159 | 0.75 | >1M | 0.87 | >1M | **0.91** | **77** |
| pulmonary_hypertension | 0.56 | >1M | 0.72 | >1M | 0.70 | >1M | **0.86** | **1461** |
| clavicular_fracture | 0.56 | >1M | 0.69 | >1M | **0.74** | 172K | 0.73 | >1M |
| esophagus_dilated | 0.45 | inf | 0.57 | >1M | 0.63 | >1M | **0.82** | **161** |
| lung_edema | 0.85 | 510 | 0.77 | 107K |  |  | **1.00** | **15** |
| diaphragms_flattened | 0.85 | 412 | 0.85 | 13228 | **0.93** | **233** | 0.90 | 272 |
| rib_fractures | 0.60 | >1M | 0.73 | >1M | **0.89** | 999 | 0.87 | **92** |
| lung_aeration | 0.67 | >1M | 0.72 | 229K | **0.96** | 49 |  |  |
| hilar_mass | 0.69 | >1M | 0.80 | 114K | **0.91** | 306 | **0.91** | **80** |
| aorta_calcification | 0.74 | >1M | 0.75 | 267K | 0.88 | 338 | **0.89** | **97** |
| mediastinum_shift | 0.68 | >1M | 0.82 | 7904 | 0.95 | **18** | **0.98** | 22 |
| lung_atelectasis | 0.64 | >1M | 0.58 | >1M | **0.89** | 7071 | 0.81 | **67** |
| pleural_air | 0.75 | >1M | 0.85 | >1M | **0.96** | **22** | 0.95 | 34 |
| cardiac_enlarged | 0.63 | >1M | 0.71 | >1M | **0.86** | 4439 | **0.86** | **2365** |

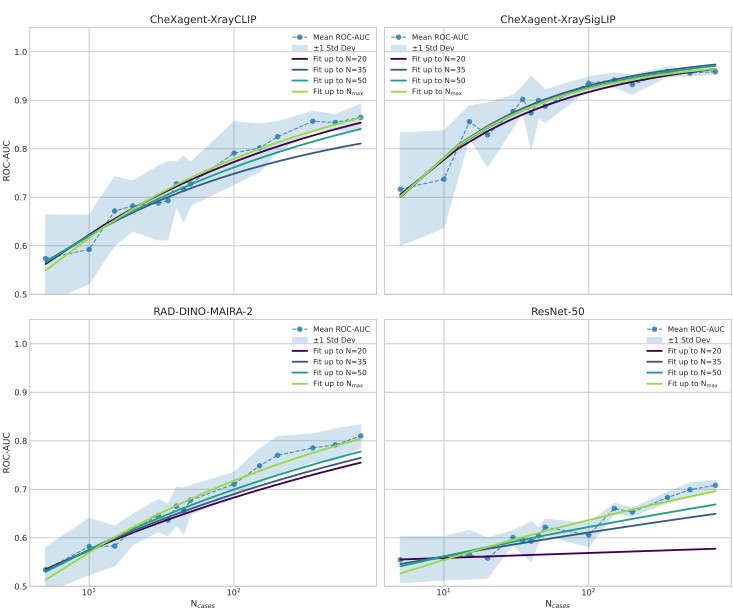

Figure 2: ROC-AUC vs the number of training examples for Lobe mass pathology.

the final achieved ROC-AUC. The slope, defined as the derivative of the fitted power-law function ($\text{ROC\_AUC}'(n) = \frac{\beta\gamma}{n^{\gamma+1}}$), was evaluated at a small sample size ($n = 5$). As shown in Figure 3, we observe a strong positive correlation that increases as more initial data points are used for fitting (*e.g.*, for ResNet-50, $r$ increases from 0.42 to 0.84 as the data cutoff for fitting is raised from 10 to 50 cases). This strong correlation provides statistical validation for our central finding: **the steepness of the initial learning curve is a reliable predictor of the model's final performance plateau.**

This allows for confident extrapolation from small, low-cost pilot experiments to estimate the full annotation budget required to meet a target performance metric.

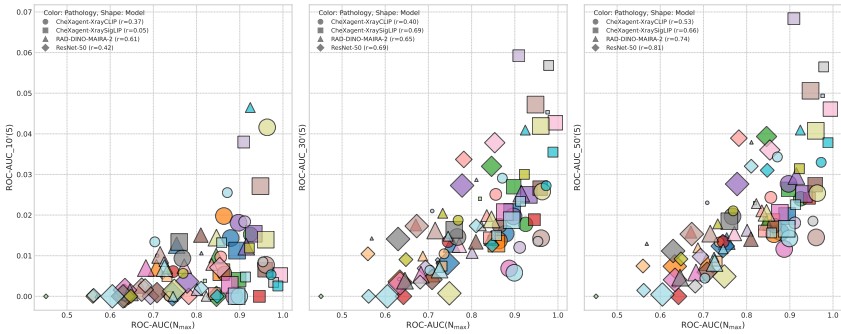

Figure 3: Correlation between the derivatives of the fitted ROC-AUC at n=5 and the value of ROC-AUC(Nmax).

### 4.4 ERROR IN PREDICTED VS. OBSERVED PLATEAU

Beyond measuring correlation, we also assessed absolute prediction error in estimating the final ROC-AUC. For each model-pathology pair, we used the power-law curve fitted at $N_{cases} = 20$ and $N_{cases} = 40$ to *extrapolate* to $N_{max}$ equal to the maximum number of examples for this pathology. We then compared this predicted $ROC\_AUC_{N_{cases}}(N_{max})$ with the actual measured value $ROC\_AUC(N_{max})$.

Figure 4 depicts how the mean absolute error (MAE) across pathologies and models evolves as we gradually increase the cutoff for fitting. Notably, the MAE decreases rapidly up to about 50-100 labeled cases, after which the benefit of additional data for partial fits diminishes.

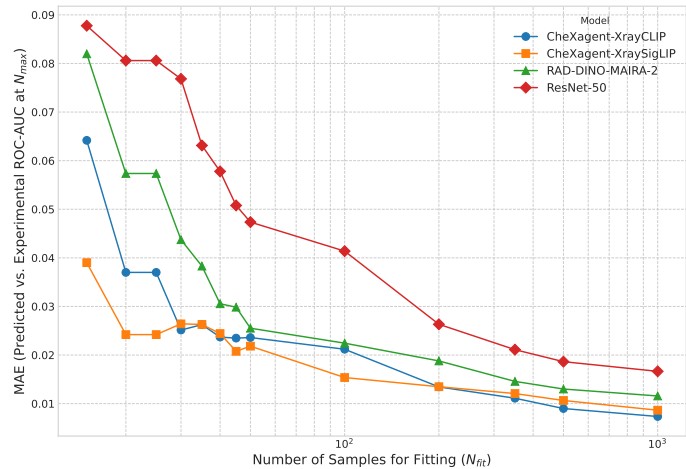

Figure 4: MAE between experimental ROC-AUC and ROC-AUC predicted on limited number of training examples.

## 5 DISCUSSION

In our experiments, we demonstrate that a straightforward process of running multiple training subsets (5 to 50 positive cases, with negative samples at a fixed ratio) allows for reliable fitting of power-law curves. By examining the initial slope and partial plateaus of these curves, we can extrapolate the performance of fine-tuned foundation models for higher training sizes. This approach is particularly useful in real-world settings where annotation is expensive, since it indicates when

additional labeling provides diminishing returns. Our findings show that, in many cases, labeling on the order of 50 to 100 positive samples per pathology is sufficient to predict—and often achieve—competitive diagnostic accuracy levels.

We acknowledge several deliberate limitations in the scope of this study. Our analysis was restricted to binary classification for 21 pathologies from a single large dataset. This focus was a methodological choice to ensure a clear, controlled, and reproducible investigation of scaling laws, avoiding the complex confounding factors of multi-class classification or dataset shift. While the principles of power-law fitting are general, future work is needed to validate the specific scaling coefficients and predictive accuracy of this method across different datasets, imaging modalities, and in multi-class scenarios.

Moreover, foundation models consistently outperform the conventional ResNet-50 baseline, underscoring not only their superior accuracy but also the improved predictability of their learning curves from limited data. Their higher initial performance effectively reduces both the total labels needed and the associated clinical costs.

The ROC-AUC threshold of 0.90 was used in this study as a consistent and illustrative benchmark. However, our methodology is fundamentally target-agnostic. Practitioners can readily substitute any clinically relevant performance threshold—be it sensitivity, specificity, or a different ROC-AUC value—to tailor the sample size estimation to their specific diagnostic needs and regulatory standards. This adaptability is key to the practical utility of our framework. We anticipate that this approach—train on subsets, fit a power law, then extrapolate to an ROC-AUC target—will inform practitioners attempting to balance annotation budgets with diagnostic performance demands when deploying chest X-ray classifiers for new pathologies.

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
