# OpenReview forum: "Sample Size Estimation for Chest X-ray Classification with Foundation Models"
_ICLR.cc/2026/Conference — Submitted to ICLR 2026_

### Official Review · Reviewer_wimB · 2025-10-15

**Soundness:** 1
**Presentation:** 1
**Contribution:** 1
**Rating:** 0
**Confidence:** 5

**Summary:**

This paper proposes a method for estimating the required sample size in CXR classification using foundation models, by fitting power-law learning curves. The authors demonstrate that with as few as 50 labeled cases, one can reliably predict AUC of models such as XrayCLIP and XraySigLIP, enabling cost-effective annotation planning.

**Strengths:**

S1: Estimating the minimum labeled data needed for fine-tuning foundation models for chest X-ray classification is of practical value.

S2: Experimental result that just 50 labeled cases can predict the model's asymptotic performance with high precision is meaningful and impressive.

**Weaknesses:**

W1: The overall content and workload are light, resembling a project report, and far from meeting top-tier conferences' acceptance standards.

W2: The introduction of power-law learning curves lacks novelty and is without adaptation to medical area.

W3: The experimental design oversimplifies clinical reality. The author enforces a fixed 1:5 positive-to-negative ratio whereas the real world disease distribution is long-tailed.

W4: The overall experimental scope is not comprehensive. For example, evaluations could also be conducted on CheXpert and ChestX-ray14, or extended to other organs.

W5: The author only tries linear probing, ignoring fine-tuning. In fact, fine-tuning is a more common practice in medical AI.

**Questions:**

See the weaknesses.

---

### Official Review · Reviewer_SL7o · 2025-10-25

**Soundness:** 2
**Presentation:** 3
**Contribution:** 2
**Rating:** 6
**Confidence:** 3

**Summary:**

The paper studies sample-size estimation for CXR classification with foundation models. They fit a power law to AUROC vs. positive-case count. They claim that early learning dynamics with as few as 50 positives accurately predict asymptotic performance and the n@90 (samples needed to reach AUROC 0.90).`

**Strengths:**

- There is a clear simple protocol with frozen encoders and a linear head. There are deterministic splits across 21 pathologies making the setup reproducible.

- The power law fitting and procedure is described in detail.

- The results table concretely illustrates n@90 reductions with foundation models.

**Weaknesses:**

- All experiments are done on a single-institution binary tasks. It is unclear whether the power-law coefficients would transfer to a dataset like CheXpert and NIH CXR14.

- The chosen model forces a monotone approach to a ceiling. This means it cannot capture peaking/double-descent, which the paper itself notes can occur in learning curves.

- The bounding of alpha [0.8, 1.0] can bias plateaus upward for difficult pathologies.

- The 90 degree rotations and large random crops are unrealistic for CXRs and might inflate sample-efficiency estimates.

- The foundation models' prior knowledge, particularly in image-text pretraining on domain reports could advantage certain pathologies.

Minor:

- l197: avalable -> available
- l196: patologies -> pathologies

**Questions:**

See weaknesses.

Additionally:

Was duplication performed after the global train/val/test split?

Do conclusions hold when unfreezing encoders? Even a small ablation on one pathology would strengthen claims.

---

### Official Review · Reviewer_USEg · 2025-10-30

**Soundness:** 1
**Presentation:** 3
**Contribution:** 1
**Rating:** 2
**Confidence:** 2

**Summary:**

The paper addresses the problem of sample size selection to reach a finding classification performance for chest X-ray imaging using some of the new representation learning mechanisms as encoders followed by a classifier. They argue that mathematical modeling (curve fitting) to scaling law curves as a function of sample size, and using the AUC as the metric to cross a certain threshold (90%), we can obtain the number of annotations needed per finding per representation method used. Since the classifier is the same for all representation learners, that is factored out and the conclusions could be reached on the basis of the representation learner.

**Strengths:**

The paper emphasizes the need for budgeting annotation as per necessary performance guarantees to reach for finding classification in chest X-rays and could be used as a practical best practice guidance for the developers of these models. The paper is bringing the methodologies used in other domains that incorporate progressive sampling by dynamically refining the power-law
estimate of the learning curve to reduce annotation overhead to the world of medical imaging. While most people use heuristics by visual observation of the ROC curves to select sample size or use incidence distribution statistics to be maintained similar in the training and testing settings, this approach is dynamically analyzing the ROC curves to set the guidance on the sample sizes per finding.

**Weaknesses:**

The biggest problem I found with this approach is its practicality. For example, our expectation already is that the number of samples for finding classification could be a function of the complexity of the finding itself (masses and nodules are the hardest to resolve in chest X-rays), and the incidence level (how often it occurs), its severity or criticality w.r.t patient outcome. We also expect that the quality of the learner matters as does the classifier architecture.
Basically the results in Table 1 are a bit all over the place indicating no conclusive guidelines. XraySigLIP takes much larger number of samples in one finding case than the for another finding, while this pattern may be opposite for another such as XrayCLIP. So no consistent pattern seen w.r.t choice of representation learner or complexity of finding.
Further, in practice, the chest X-ray read requires simultaneous detection of multiple findings in chest X-rays. So if a particular finding needs more images for training, while another common condition can be detected with fewer samples, then the higher complexity finding sample size decisions will dominate, meaning we may end up annotating more images anyway. Most generative AI models train on a multi-label classifier model, and this study may be better done in those scenarios.

**Questions:**

How does the result and learning generalize, are 50 images sufficient for any stable finding?
It is also a common observation that the performance can degrade with more training data as more variety is seen. Have you tried looking at this aspect?
Why not sample following the incidence distribution of the findings in the population in a certain demographics?

Finally, was it really necessary to do scaling law curve modeling to reach these conclusions?

---

### Official Review · Reviewer_W8eQ · 2025-11-09

**Soundness:** 2
**Presentation:** 3
**Contribution:** 1
**Rating:** 0
**Confidence:** 5

**Summary:**

This paper studies finetuning scaling laws for xray foundation models to better understand how well the minimum number of finetuning examples required for a given xray foundation model can be predicted; tested on MIMIC-CXR.

**Strengths:**

The paper is well written and simple to follow.

**Weaknesses:**

Unfortunately, I am strongly advocating for rejection here, as the level of contribution conducted in this paper is minimal:
* Scaling laws (particular those of power law nature) are well established (as the authors also correctly reference); both for model pretraining, as well as test-time scaling for all forms of models and training approaches in settings much more general than those studied in this paper. The authors effectively conduct a small-scale scaling law study on a limit range of models using existing power law formulations, just specifically applied to the domain of medical Xray vision model finetuning.
* To me, it is hard to see any novelty or immediate relevant contribution - the fact that these power laws exist is well known across domains, and the scaling studies conduct (on both a dataset and model level) are too limited for the fitted laws to be generally applicable.

**Questions:**

See Weaknesses.

---

### Meta-Review · Area_Chair_zLna · 2026-01-04

**Summary:**

Paper received 2 strong rejects and 1 reject and did not submit a rebuttal.

**Reviewer Concerns:**

The reviewers were concerned about the novelty, experimentation, and practicality.

**Reviewer Scores:**

There was no rebuttal so the scores would not have changed. Due to the lack of rebuttal, this paper warrants rejection.

---

### Decision · Program_Chairs · 2026-01-26

Reject